# Ultraviolet Photodetector Based on a Beta-Gallium Oxide/Nickel Oxide/Beta-Gallium Oxide Heterojunction Structure

**DOI:** 10.3390/s23198332

**Published:** 2023-10-09

**Authors:** Shinji Nakagomi

**Affiliations:** Department of Information Technology and Electronics, Faculty of Science and Engineering, Ishinomaki Senshu University, Ishinomaki 986-8580, Japan; nakagomi@isenshu-u.ac.jp

**Keywords:** β-Ga_2_O_3_, NiO, UV, photodetector, amplification

## Abstract

In this paper, an n–p–n structure based on a β-Ga_2_O_3_/NiO/β-Ga_2_O_3_ junction was fabricated. The device based on the β-Ga_2_O_3_/NiO/β-Ga_2_O_3_ structure, as an ultraviolet (UV) photodetector, was compared with a p–n diode based on a NiO/β-Ga_2_O_3_ structure, where it showed rectification and 10 times greater responsivity and amplified the photocurrent. The reverse current increased in proportion to the 1.5 power of UV light intensity. The photocurrent amplification was related to the accumulation of holes in the NiO layer given by the heterobarrier for holes from the NiO layer to the β-Ga_2_O_3_ layer. Moreover, the device could respond to an optical pulse of less than a few microseconds.

## 1. Introduction

Gallium oxide (Ga_2_O_3_) has attracted attention recently as a next-generation power semiconductor material. Among the several poly-types of Ga_2_O_3_, β-Ga_2_O_3_ with a bandgap of 4.9 eV has the most thermodynamically stable crystal structure. Due to its optimal physical properties, several power devices and ultraviolet (UV) photodetectors have been studied [1,2,3,4,5]. However, it is difficult to obtain p-type conduction in β-Ga_2_O_3_. Therefore, devices based on β-Ga_2_O_3_ are currently restricted to Schottky and metal oxide semiconductor devices, such as Schottky diodes and field-effect transistors, respectively.

As a p-type metal oxide semiconductor, NiO, CuGaO_2_, Cu_2_O, and NiCo_2_O_4_, which have band gaps of 3.7 eV, 3.6 eV, 2.4 eV, and 2.1 eV, respectively, have been studied. NiO has the widest band gap among them. Kokubun et al. [6] proposed a NiO/β-Ga_2_O_3_ p–n heterojunction diode using the rare metal oxide NiO, which has p-type conduction and demonstrated good rectification of the diode. The same research group suggested that NiO was suitable for β-Ga_2_O_3_ in crystal orientation [7]. Since then, various diodes based on the NiO/β-Ga_2_O_3_ p–n heterojunction have been studied for application in power devices [8,9] or UV photodetectors [10,11,12,13].

UV light rays pose serious health problems for humans, such as damage to the skin, skin cancer, and white cataracts. UV light is classified as UV-A, -B, and -C, corresponding to wavelengths of 315–400 nm, 280–315 nm, and 100–280 nm, respectively. Because UV-A and -B cause light damage to the skin, it is important to be vigilant for UV light illumination. A UV-C photodetector is required for concerns of UV radiation through the holes in the ozone layer. UV-C light is also used for sterilization purposes. In order to protect against UV light, proper UV photodetectors are required. Therefore, many types of UV photodetectors have been studied [14,15,16,17,18,19,20]. There are several approaches, such as amorphous film, nano-rods, and combinations with other materials.

The wavelength of the photodetector relates to the bandgap of the active layer. If a photodetector for UV-B or -C is needed, a bandgap semiconductor wider than 3.9 eV should be used. Al_X_Ga_1-X_N and Ga_2_O_3_ are typical candidates. Because a larger aluminum mole fraction of Al_X_Ga_1-X_N is required for a wider bandgap, the difficulties inherent in growing crystals are increased. Meanwhile, β-Ga_2_O_3_ is a promising candidate for a UV-C photodetector because β-Ga_2_O_3_ is chemically stable, and commercial substrates are available. It is also expected that it can be used as a solar-blind UV photodetector.

Several UV photodetectors based on Ga_2_O_3_ have been studied. The resistance type [10] is based on the mechanism of the photoconductor. In the diode type, short-circuit current or reverse current [11,12,13] is used as the sensor signal. In the diodes based on the NiO/β-Ga_2_O_3_ p–n heterojunction, there are some difficult assignments relating to the hetero growth and interface trap due to the heterojunction. Although the author has studied the crystal orientation relationship between NiO and β-Ga_2_O_3_, there remain issues to be resolved.

In UV photodetectors, low dark current, responsivity, and response times are important factors. Several device structures have been studied to achieve these requirements. However, the amplification of photocurrent has not been reported in devices based on β-Ga_2_O_3_. A new device is required to obtain this amplification of the photocurrent. Therefore, in this study, the amplification of photocurrent by fabricating a β-Ga_2_O_3_/NiO/β-Ga_2_O_3_ structure is described. The device, based on a β-Ga_2_O_3_/NiO/β-Ga_2_O_3_ structure, has never been reported. The author studied the device as a UV photodetector. The photodetector exhibited 10 times greater responsivity and amplified the photocurrent. Moreover, a response with a high speed of the order of microseconds was demonstrated using pulse-driven UV-LED. The photodetection and amplification mechanism is discussed using a schematic band model.

## 2. Experimental

### 2.1. Device Structure and Fabrication

The devices were fabricated using purchased β-Ga_2_O_3_ substrates through a thin-film formation technique by the sol–gel method. Lift-off of the SiO_2_ sacrificial layer, photolithography, and formation of metal electrodes were also performed.

Figure 1 shows the cross-sectional structure of the device and a photograph of its top view. The (001) β-Ga_2_O_3_ epitaxial substrates were used and purchased from Novel Crystal Technology Inc. The substrates were Sn-doped n-type substrates with a carrier density of 5 × 10^18^ cm^−3^ and had a Si-doped 8-µm epitaxial layer with a carrier density of 3 × 10^16^ cm^−3^. A 100 nm-thick film of the 10% Li-doped NiO was selectively formed on the β-Ga_2_O_3_ substrate. Afterward, a 100 nm-thick layer of the undoped β-Ga_2_O_3_ was selectively formed on the NiO layer. In both cases, the sol–gel method was used, and the SiO_2_ sacrificial layer was removed using the lift-off process.

A SiO_2_ thin film was formed on the β-Ga_2_O_3_ substrate through spin coating of a SiO_2_ solution and annealing. The SiO_2_ thin film was etched to make a hole for the selective formation of the NiO layer after patterning by photolithography, where the Li-doped NiO layer was formed using the sol–gel method. The 10% Li was added as a solution for the Ni. The resistivity of the 10% Li-doped NiO film was about 0.5 Ω·cm. The concentration and mobility of holes in the doped NiO were ~3 × 10^20^ cm^−3^ and 0.05 cm^2^/Vs, respectively [6,21]. The absorption properties of the Li-doped NiO film are shown in our previous work [21].

Once again, a SiO_2_ thin film was formed on the sample. The SiO_2_ thin film was etched through the same lithographic process, and a smaller hole was formed. Then, an undoped β-Ga_2_O_3_ layer was formed on the NiO layer through the sol–gel method using a gallium isopropoxide solution [22]. The condition of the heat treatment was 750 °C 1 h. In our recent experiment, the resistivity of β-Ga_2_O_3_ formed on the MgO substrate, which has the same cubic structure and lattice constant as NiO, was ~150 Ω·cm. The absorption properties of the β-Ga_2_O_3_ film prepared by the sol–gel method are shown in our previous work [22]. For information purposes, the transmission spectra of the Li-doped NiO and β-Ga_2_O_3_ films formed on the sapphire substrate are shown in Appendix A.

The SiO_2_ layers were chemically etched using an HF solution to remove the NiO and β-Ga_2_O_3_ layers formed on the SiO_2_ layer. The NiO and β-Ga_2_O_3_ layers on the β-Ga_2_O_3_ substrate were selectively retained, as shown in Figure 1.

Ohmic Ti/Al/Pt/Au electrodes were selectively formed on the top of the β-Ga_2_O_3_ layer and the bottom of the substrate through an annealing process at 500 ℃. The bottom electrode on the β-Ga_2_O_3_ substrate had a hole positioned underneath the top electrode for illumination of UV light. The diameter of the hole was around 1 mm. Finally, an Au electrode was selectively formed on the NiO layer. The diameter of the top electrode on the β-Ga_2_O_3_ layer was 0.5 mm, the diameter of the β-Ga_2_O_3_ layer was 0.7 mm, and the area of the NiO layer was approximately 1 mm^2^.

The X-ray diffraction patterns (2θ–ω scan) of the samples were measured before all the metal electrodes were formed. The vertical line was on a logarithmic scale. Because the (001) β-Ga_2_O_3_ substrate was used, strong (001)-related reflection peaks were observed in the patterns, as shown in Figure 2. The peak from the NiO layer was a weak reflection of the (133) plane of NiO. The crystal orientation of the NiO thin film formed on the (001) β-Ga_2_O_3_ substrate was reported in a previous work. The (133)-oriented NiO layer was formed on the (001) β-Ga_2_O_3_ substrate. However, because the (133) plane was slightly inclined to the (001) plane of β-Ga_2_O_3_, the NiO (133)-related reflection peak was weakly observed. The reflection peaks of β-Ga_2_O_3_ were not observed except for the (001)-related diffraction. This suggests that the top β-Ga_2_O_3_ layer formed on the NiO layer is oriented to (001), the same as the (001) β-Ga_2_O_3_ substrate. The crystal orientation of the β-Ga_2_O_3_ thin film formed on NiO was studied using a MgO substrate, which has the same cubic structure and a similar lattice constant.

Figure 3 illustrates the scanning electron microscope (SEM) image of the β-Ga_2_O_3_ surface and NiO layers formed on the (001) β-Ga_2_O_3_ substrate. The top β-Ga_2_O_3_ layer was selectively formed by a lift-off process employing SiO_2_ on the NiO layer, as described above. The NiO layer comprised small crystal grains, which maintained the crystal orientation. The top β-Ga_2_O_3_ layer on NiO comprised smaller crystal grains.

The top electrode on the β-Ga_2_O_3_ layer, the Au electrode on the NiO layer, and the bottom electrode on the β-Ga_2_O_3_ substrate are represented by E, B, and C, respectively, as shown in Figure 1. The structure between B and C corresponds to a p–n junction based on NiO/β-Ga_2_O_3_, which has a composition similar to that reported in [6]. The structure between E and C corresponds to an n–p–n junction based on β-Ga_2_O_3_/NiO/β-Ga_2_O_3_, which is a novel structure. The structure between E and C was compared with that between B and C.

In this study, a special electrode structure was not fabricated on the device so that the UV light could easily pass through the top electrode. Therefore, when the device was illuminated in the direction of the top electrode, the UV light that reached the junction region was weak because the top electrode may shut out the UV light. On the other hand, in the case when the device was illuminated on the bottom side, the wavelength region corresponding to the fundamental absorption of the β-Ga_2_O_3_ substrate is somewhat absorbed before the UV light reaches the junction region and the NiO layer. The UV light with a wavelength corresponding to the fundamental absorption of the NiO can reach the junction region by the window effect of β-Ga_2_O_3_ with a wider bandgap.

### 2.2. Measurements

The current–voltage characteristics were measured using a source-measure unit (Keithly 6487). UV light was irradiated from 0.1% to 100% using a deuterium (D2) lamp through several neutral-density (ND) filters. The light power density of the deuterium lamp was 22 mW/cm^2^ as a rough estimate using a standard photodiode. The distance between the lamp and the bottom of the substrate was around 150 mm. To obtain the responsivity spectrum, a Xenon arc lamp was used with a monochromator as the optical excitation source. The wavelength was varied from 200 to 450 nm with an increment of 5 nm. The photoresponse spectra were compared based on the photoresponse of a calibrated photodiode.

A UV-LED with a peak wavelength of 310 nm was used as an optical source to measure the transient response curves. A pulse voltage was applied to the UV-LED, and the optical output was monitored using a Si avalanche photodiode (APD). UV light was irradiated on the bottom side of the UV photodetector with a β-Ga_2_O_3_/NiO/β-Ga_2_O_3_ through a quartz lens. The current response pulse of the detector was amplified and monitored using a digital oscilloscope.

## 3. Results and Discussion

### 3.1. Current–Voltage Characteristics

Figure 4a shows the current–voltage characteristics between the B and C electrodes corresponding to a p–n junction based on NiO/β-Ga_2_O_3_ structure. The current increased exponentially when the B electrode on the p-type NiO layer was positively biased. Under the reverse bias condition, the current was maintained at nearly 0.1 nA in the dark. The rectification ratio of the diode, which was calculated from the current at 5 V and −10 V bias, was about 1.8 × 10^7^. This rectifying property has been reported in a previous study [6]. Under the UV light illumination, the reverse current increased, whereas the forward current slightly increased in the low bias region. The rectification ratio of the diode, which was calculated from the current at 5 V and −10 V bias, was about 2.7 × 10^4^. This behavior between the B and C electrodes for UV light illumination is similar to the photodetection of a conventional p–n junction.

Figure 4b illustrates the current–voltage characteristics between the E and C electrodes corresponding to an n–p–n junction based on β-Ga_2_O_3_/NiO/β-Ga_2_O_3_ structure. The forward current between E and C was lower than that between B and C, suggesting a higher series resistance between E and C than that between B and C. Under the reverse bias condition, the current was maintained at 0.1 nA in the dark, and the reverse current increased; furthermore, the reverse current increased with increasing bias voltage under UV light illumination. This behavior is different from the B and C electrodes of the p–n junction based on NiO/β-Ga_2_O_3_. The rectification ratio of the diode, which was calculated from the current at 5 V and −10 V bias, was about 2.8 × 10^4^ and 35 in the dark and under UV light, respectively.

Figure 5a,b illustrate the dependence of reverse current–voltage characteristics up to 100 V of the p–n junction based on the NiO/β-Ga_2_O_3_ structure and the device based on the β-Ga_2_O_3_/NiO/β-Ga_2_O_3_structure on the relative intensity of UV light illumination, respectively. The intensity of UV light illumination was varied using a D_2_ lamp and several ND filters. The dependence of the current–voltage characteristics of B–C on the relative intensity of UV light illumination is also shown in Figure 5a. With an increase in the intensity of UV light illumination, the reverse current was increased, maintaining the saturation property. The UV light on–off ratio calculated from the current at 100 V was about 628.

Figure 5b also shows the dependence of the current–voltage characteristics of E–C on the relative intensity of UV light illumination. The characteristics of E–C were similar to the characteristics of B–C at an intensity of UV light illumination lower than 1%. However, with the increasing intensity of UV light illumination, the current did not maintain the saturation property and increased with an increase in the bias voltage. It was observed that the electrical resistance between E and C decreased with an increase in the relative intensity of UV light illumination. The reverse current of E–C at 100 V increased more than 10 times compared to the current of B–C under the condition of 100% UV light illumination. The UV light on–off ratio calculated from the current at 100 V was about 10130.

Figure 6 shows the relationship between the reverse current of the device biased at 100 V and the relative intensity of UV light illumination. Note that logarithmic scales were employed on both the horizontal and vertical axes. The relationship for B–C, which corresponds to the p–n junction based on the NiO/β-Ga_2_O_3_ structure, had a linear inclination of 0.84. In contrast, the relationship for E–C, which corresponds to the n–p–n device based on the β-Ga_2_O_3_/NiO/β-Ga_2_O_3_ structure, had a larger inclination of 1.5. The reverse current increases in proportion to the 1.5 power of UV light intensity. This means that there is photocurrent amplification in the device with the innovation of the β-Ga_2_O_3_/NiO/β-Ga_2_O_3_ structure. This is the first demonstration of photocurrent amplification in a UV photodetector based on a β-Ga_2_O_3_ or a β-Ga_2_O_3_/NiO junction.

### 3.2. Photo Responsivity

Figure 7a shows the responsivity spectrum of B–C, which corresponds to the p–n junction based on the NiO/β-Ga_2_O_3_ structure. The highest sensitivity was observed at 285 nm. With increasing reverse bias voltage from 10 to 100 V, the responsivity was increased at the same wavelength as the maximum responsivity.

Figure 7b shows the responsivity spectrum of E–C, which corresponds to the n–p–n device based on the β-Ga_2_O_3_/NiO/β-Ga_2_O_3_ structure. When the device was biased at 10 V, the responsivity spectrum, whose peak was at about 290 nm, was broad, and the maximum responsivity was 290 nm, similar to that of the p–n diode shown in Figure 7a. It should be noted that the scale of the vertical line in Figure 7b is about 10 times larger than that in Figure 7a. With an increase in bias voltage, the wavelength of the maximum responsivity moved to 275 nm, and the maximum responsivity increased up to 10 mA/W when the bias was 100 V. The maximum responsivity was about 10 times greater than the responsivity of B–C. When the maximum responsivity was at about 290 nm under the bias condition of 10 V, it is suggested that the generated carriers in the NiO layer contributed to the photocurrent. When the maximum responsivity was suddenly increased at ~275 nm under the bias voltage >40 V, the generated charge carriers near the interface between NiO and β-Ga_2_O_3_ may have contributed to the photocurrent. The generated holes flow to the NiO layer and accumulate in the NiO layer. They contribute to the amplification of the electron flow from the top β-Ga_2_O_3_ layer to the NiO layer.

In this study, the obtained responsivity was compared with that of several UV photodetectors based on Ga_2_O_3_. Table 1 shows the comparison of structural and photo responsivity. The highest photoresponsivity of 1720.2 A/W was achieved in the ITO/β-Ga_2_O_3_ structure [16]. Furthermore, in the transparent amorphous Ga_2_O_3_ structure, a high responsivity of 2.66 A/W and 5.78 A/W was obtained [15,16]. This suggests that a transparent electrode structure effectively enhances photo responsivity.

Several studies have investigated the UV photodetectors with a NiO/β-Ga_2_O_3_ structure [11,12,13]. Although the responsivity was low at the beginning, it was enhanced, and a higher responsivity of 27.43 A/W was achieved for the device based on an ITO/NiO/β-Ga_2_O_3_ structure [11]. The photodetector that used both nanowire and CH_3_NH_3_PbI_3_ achieved a higher responsivity of 254 mA/W [17]. The responsivity of 10 mA/W obtained in this study is lower than other detectors reported in the literature [10,11]. However, the presented detector demonstrated the amplification of photocurrent for the first time. This is a remarkable point.

### 3.3. Transient Response

The transient responses of the device to an optical pulse were measured using a UV-LED whose peak wavelength of light emission was 310 nm. The UV light can reach the junction region by passing through the β-Ga_2_O_3_ substrate without absorption by the window effect, even under the UV light illumination from the substrate’s bottom. The UV-LED was operated by a pulse generator. Figure 8a shows the waveform of the photoemission from the UV illumination measured by using an avalanche photodiode module and a digital oscilloscope. The optical pulse width and repetition period were 2 and 15 µs, respectively.

Figure 8b shows the photoresponse curves of E–C and B–C electrodes reversely biased at 10 V. The E–C and B–C correspond to the device based on the β-Ga_2_O_3_/NiO/β-Ga_2_O_3_ and the NiO/β-Ga_2_O_3_ (B–C) structures, respectively. The current with a road resistance of 50 kΩ was amplified and recorded in the digital oscilloscope. Clear sensing responses to the UV light pulses of 2 µs in width were observed for both E–C and B–C. The response level of E–C was higher than that of B–C. It was demonstrated that the device was able to respond to the optical pulse for shorter than a few microseconds. This response time was rather quick compared to the results of milliseconds reported in [5]. When the device’s properties and the circuit system to detect current pulses are improved, the device is expected to respond at a higher speed.

## 4. Discussion

As described by the characteristics shown in Figure 4b, Figure 5b, Figure 6 and Figure 7b, the presented device based on the β-Ga_2_O_3_/NiO/β-Ga_2_O_3_ structure has the function of amplification contrary to the p–n diode based on the NiO/β-Ga_2_O_3_ structure. It is certain that the addition of the β-Ga_2_O_3_ top layer caused the amplification of the photocurrent. To study the amplification mechanism of the detector with a β-Ga_2_O_3_/NiO/β-Ga_2_O_3_ structure, we conducted several experiments and used a schematic band model.

Normalized photoresponsivities of several diodes based on β-Ga_2_O_3_ were compared using the UV photodetector with a β-Ga_2_O_3_/NiO/β-Ga_2_O_3_ structure. The photoresponsivity of the β-Ga_2_O_3_ Schottky diode with a thin Au electrode is shown in Figure 9a, with the structure shown in (a1). The highest responsivity was obtained at 225 nm, which corresponds to the fundamental absorption of β-Ga_2_O_3_.

The normalized photoresponsivities of the diode based on a NiO/β-Ga_2_O_3_ hetero p–n diode are shown in Figure 9b,c, with the structures shown in (b1) and (c1). In the case of (b), UV light was illuminated on the diode from the surface of the thin Au electrode on the NiO layer. The UV light could reach the NiO layer through the thin Au electrode. The highest responsivity was obtained at 335 nm, which is similar to that obtained in the case where UV light was illuminated on the E–C detector from the surface side of the top electrode, as described above. In the case of (c), UV light was illuminated on the NiO/β-Ga_2_O_3_ hetero p–n diode from the bottom side of the β-Ga_2_O_3_ substrate, and the highest responsivity was obtained at 285 nm.

Figure 9d,e show the normalized photoresponsivities of the detector with NiO/β-Ga_2_O_3_ and β-Ga_2_O_3_/NiO/β-Ga_2_O_3_ structures. These responsivity spectra were obtained from the properties of the B–C and E–C electrodes biased at −100 V, respectively. The highest responsivities were obtained at 285 and 275 nm, respectively, and the wavelength of 285 nm is the same as that in Figure 9c. Because UV light was irradiated on the diode from the bottom of the 0.5-mm-thick Ga_2_O_3_ substrate, most of the UV light with shorter wavelengths corresponding to the fundamental absorption of β-Ga_2_O_3_ became weak when the light reached the region near the NiO/Ga_2_O_3_ junction. Though the UV light with shorter wavelengths was weak, the photoresponse at 275 nm, in other words, the photoresponse for higher-energy UV light with wavelengths shorter than 285 nm, increased quickly in the E–C detector. This indicates that the generated carriers near the interface between NiO and β-Ga_2_O_3_ layers_,_ which has a larger bandgap than NiO, contributed to the photocurrent. NiGa_2_O_4_ was synthesized at the interface between NiO and β-Ga_2_O_3_ under high-temperature conditions. The NiGa_2_O_4_ film has a bandgap value between that of NiO and β-Ga_2_O_3_. The holes generated in the β-Ga_2_O_3_ layer or Ga_2_O_3_ substrate may contribute to the photoresponse because the Ga_2_O_3_ layer on the NiO layer acts as a barrier for holes to flow toward the surface.

To elucidate the overview, a schematic band diagram of the device based on the β-Ga_2_O_3_/NiO/β-Ga_2_O_3_ structure is shown in Figure 10. In the p–n heterojunction comprising p-type NiO and n-type β-Ga_2_O_3_, the energy barrier for the flow from NiO to β-Ga_2_O_3_ for the holes in the NiO layer is greater than the energy barrier for the flow from β-Ga_2_O_3_ to NiO for the electrons in the β-Ga_2_O_3_ layer. The difference in the energy barrier was reported in a previous study [6]. The band offsets of ΔEc and ΔEv in the heterojunction between NiO and β-Ga_2_O_3_ were 2.2 and 3.4 eV, respectively.

Recombination centers exist at the interface between the NiO layer and the β-Ga_2_O_3_ substrate because of the lattice mismatch. The current in the NiO/β-Ga_2_O_3_ heterojunction diode is largely attributed to interface recombination. Furthermore, in the device based on the β-Ga_2_O_3_/NiO/β-Ga_2_O_3_ structure, the electrons in β-Ga_2_O_3_ may recombine with the holes in the NiO layer through the interface between the β-Ga_2_O_3_ and NiO layers.

UV light with a wavelength >260 nm reaches the NiO layer without attenuation, even when the UV light is illuminated from the bottom of the β-Ga_2_O_3_ substrate. When the device was illuminated with UV light and electron–hole pairs were generated on the NiO layer, the holes generated by the absorption of UV light in the NiO layer, including the holes that flowed to the NiO layer from the depletion layer of the p–n junction of B–C or B–E, are expected to accumulate in the NiO layer. The accumulated holes act as positive charges and positively bias the p–n heterojunction of B–E. Thus, the barrier height for electrons in the β-Ga_2_O_3_ layer mentioned above is decreased, which increases the electron flow from the β-Ga_2_O_3_ layer to the NiO layer. It is also expected that the accumulated holes decrease the electrical resistance of the NiO layer. This is an anticipated mechanism to amplify the photocurrent. In the p–n diode based on the NiO/β-Ga_2_O_3_ structure, the holes generated by the absorption of UV light in the depletion layer of the p–n junction of B–C reversely biased may flow to the NiO layer but will not accumulate in the NiO layer. Therefore, amplification of the photocurrent does not occur in the p–n diode structure.

## 5. Conclusions

The n–p–n structure was constructed by fabricating both the NiO layer formed on the β-Ga_2_O_3_ substrate and the β-Ga_2_O_3_ layer formed on the NiO layer. The device based on the β-Ga_2_O_3_/NiO/β-Ga_2_O_3_ (n–p–n) structure, as a UV photodetector, was compared with the p–n diode based on the NiO/β-Ga_2_O_3_ structure. The device based on the β-Ga_2_O_3_/NiO/β-Ga_2_O_3_ structure showed a 10 times greater responsivity and amplified the photocurrent. This amplification of the photocurrent is the first in a UV photodetector based on NiO and β-Ga_2_O_3_. The photocurrent of the device increased in proportion to the 1.5 power of relative UV light intensity. The photocurrent amplification is related to the accumulation of holes in the NiO layer provided by the heterobarrier for holes from the NiO layer to the β-Ga_2_O_3_ layer. Moreover, the device could respond to an optical pulse of less than a few microseconds.

The amplification of photocurrent is an important achievement. However, the photoresponsivity of the detector based on the β-Ga_2_O_3_/NiO/β-Ga_2_O_3_ structure is not high at this stage. In future studies, a structure that enables UV light to reach the junction region should be developed to improve the responsivity of the detector.

## Figures and Tables

**Figure 1 sensors-23-08332-f001:**
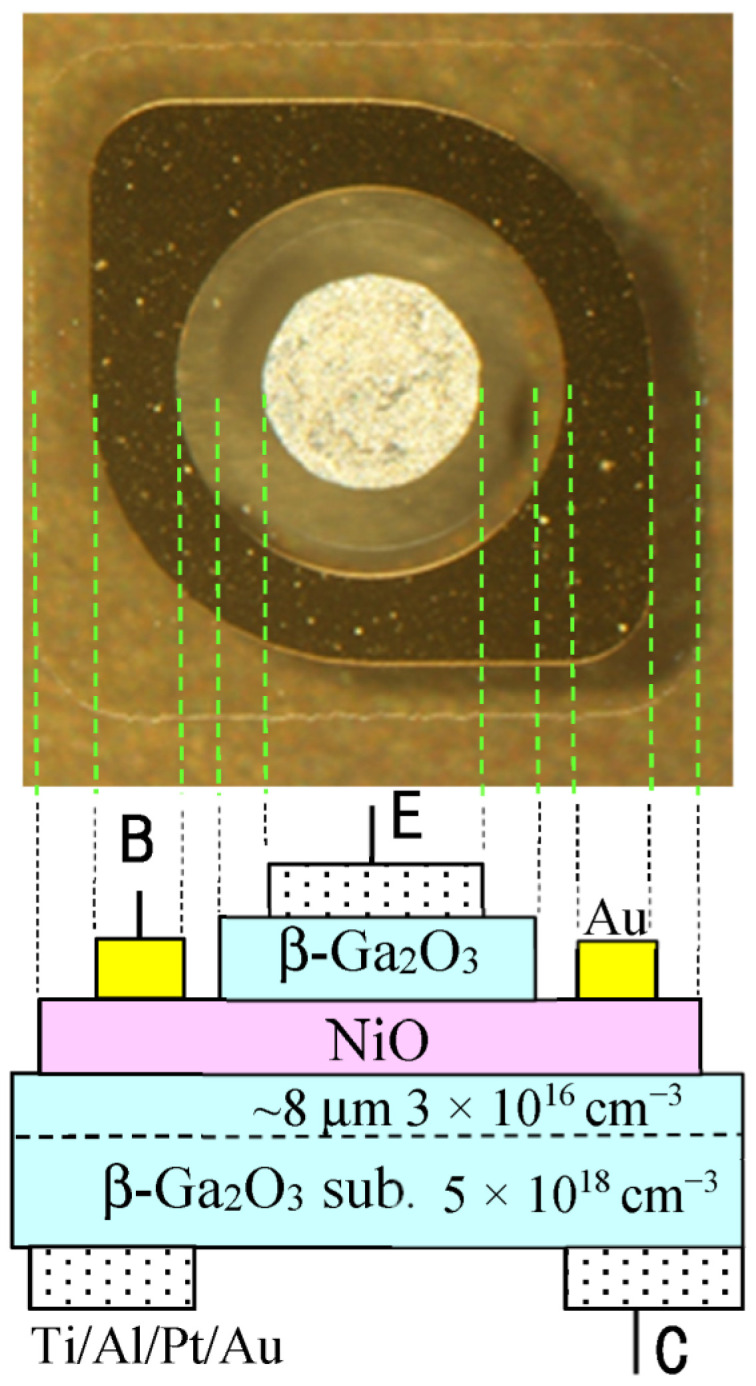
Cross-sectional structure of the device and a photograph of its top view. Three electrodes are represented by E, B, and C.

**Figure 2 sensors-23-08332-f002:**
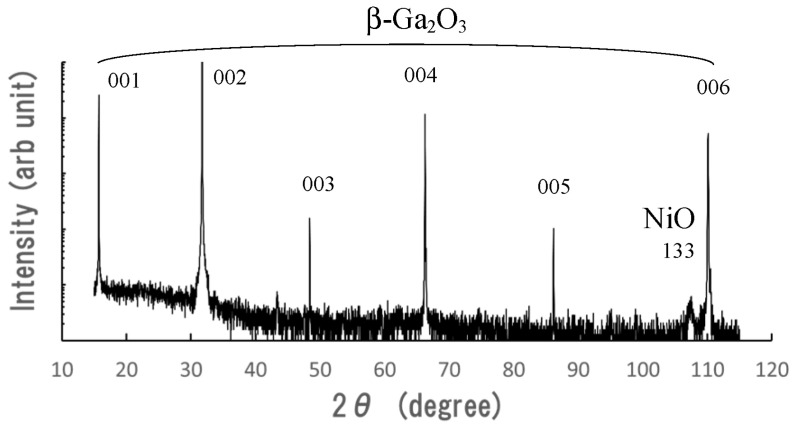
X-ray diffraction patterns (2θ–ω scan) of the β-Ga_2_O_3_ and NiO layers formed on the (001) β-Ga_2_O_3_ substrate.

**Figure 3 sensors-23-08332-f003:**
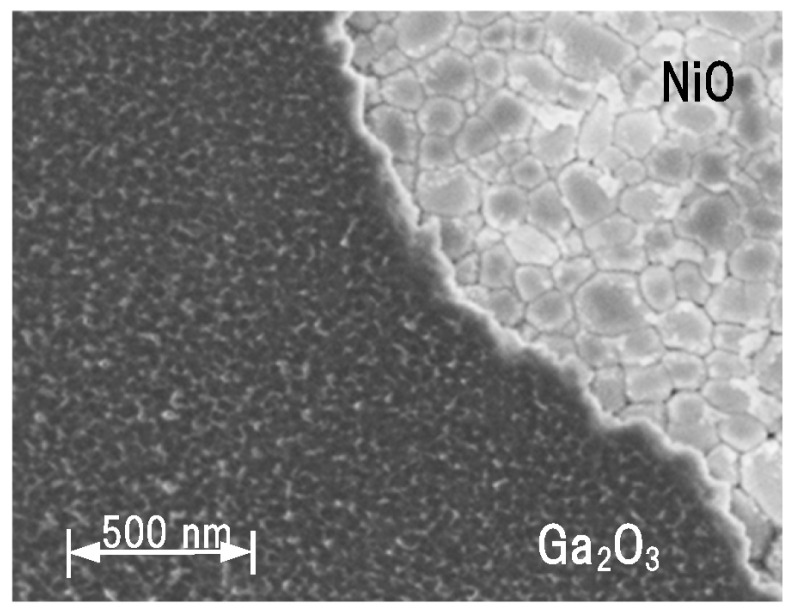
SEM image of the β-Ga_2_O_3_ surface and NiO layers formed on the (001) β-Ga_2_O_3_ substrate.

**Figure 4 sensors-23-08332-f004:**
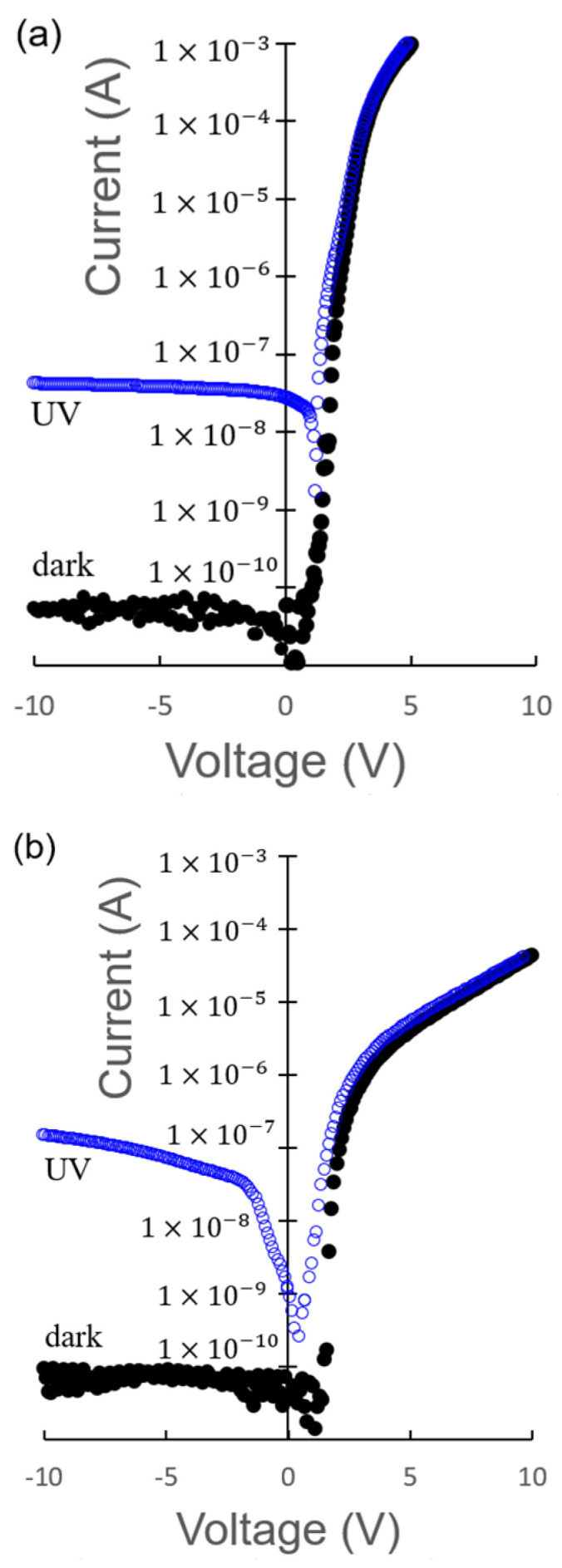
Current–voltage characteristics of (**a**) the p–n junction based on the NiO/β-Ga_2_O_3_ structure and (**b**) the device based on the β-Ga_2_O_3_/NiO/β-Ga_2_O_3_ structure under dark and UV light illumination conditions.

**Figure 5 sensors-23-08332-f005:**
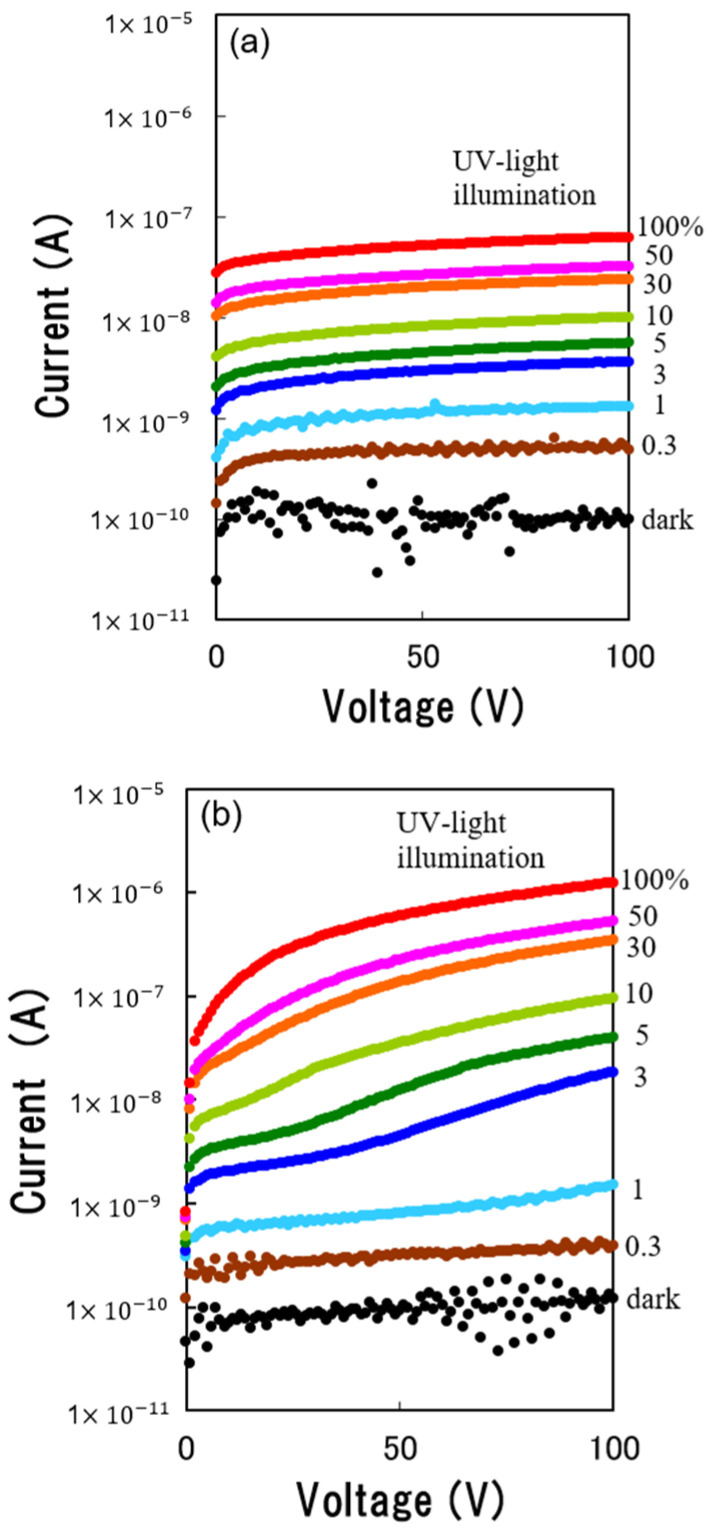
Dependence of reverse current–voltage characteristics of (**a**) the p–n junction based on the NiO/β-Ga_2_O_3_ structure and (**b**) the device based on the β-Ga_2_O_3_/NiO/β-Ga_2_O_3_ structure on the relative intensity of UV light illumination.

**Figure 6 sensors-23-08332-f006:**
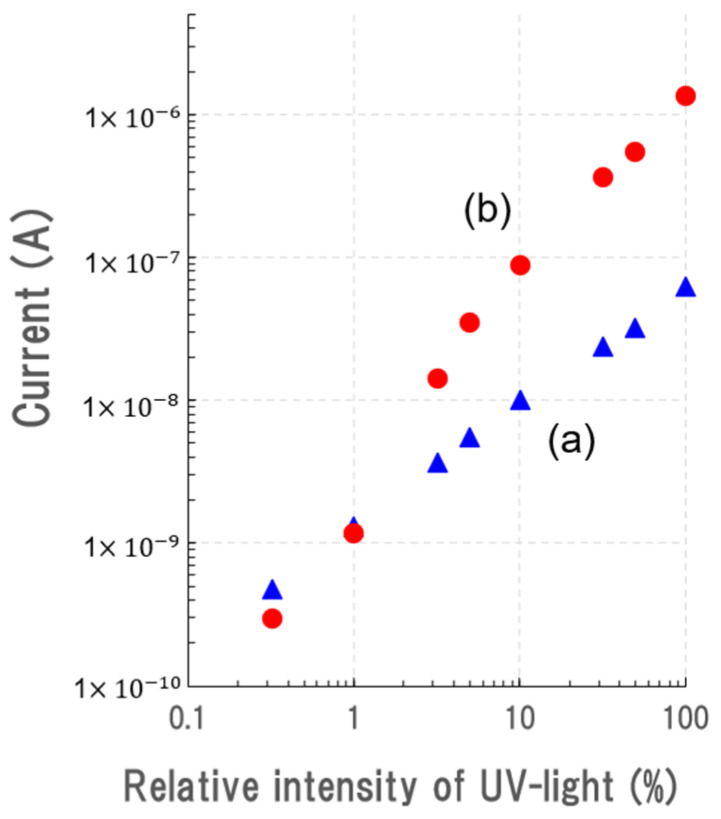
Relationship between reverse current at 100 V and the relative intensity of UV light illumination of (a) the p–n junction based on the NiO/β-Ga_2_O_3_ structure and (b) the n–p–n device based on the β-Ga_2_O_3_/NiO/β-Ga_2_O_3_ structure.

**Figure 7 sensors-23-08332-f007:**
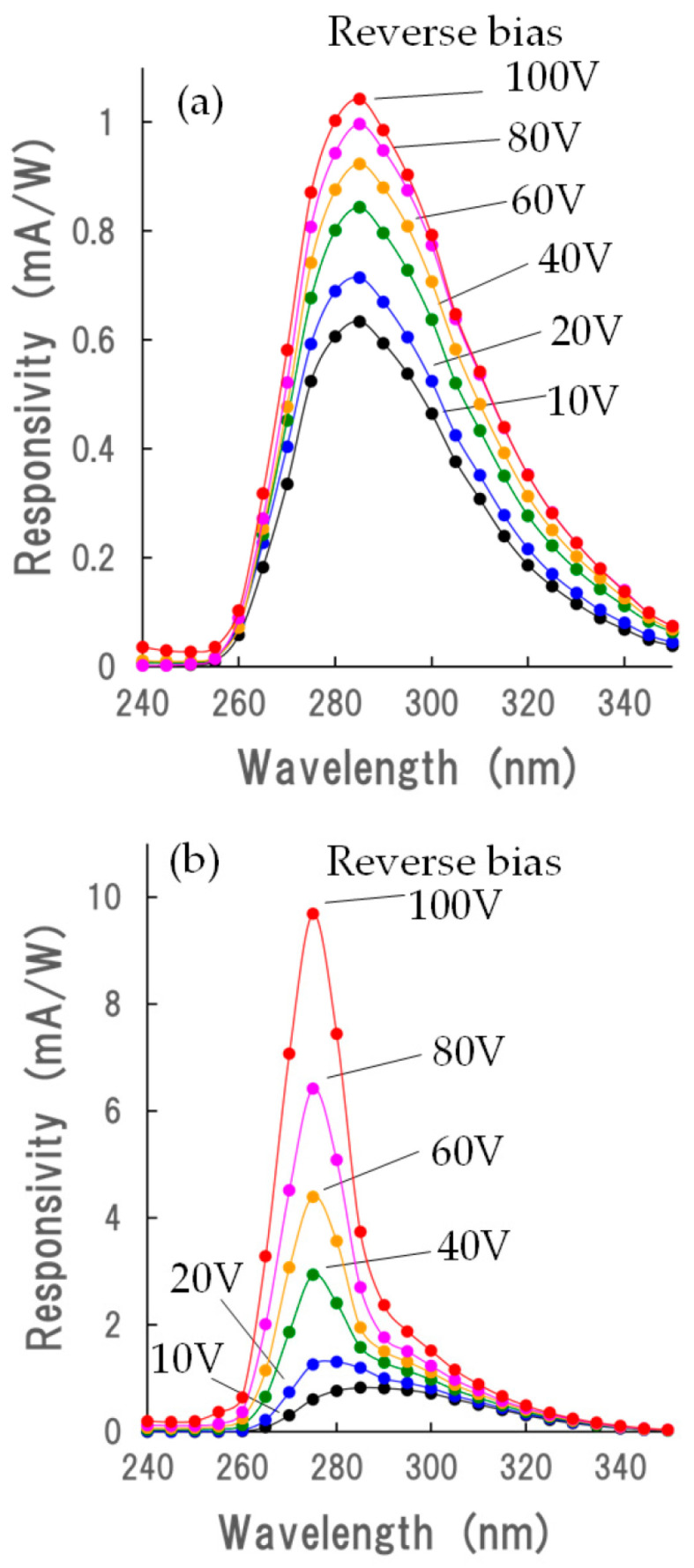
Dependence of responsivity spectrum on reverse bias voltage of (**a**) the p–n junction based on the NiO/β-Ga_2_O_3_ structure and (**b**) the device based on the β-Ga_2_O_3_/NiO/β-Ga_2_O_3_ structure.

**Figure 8 sensors-23-08332-f008:**
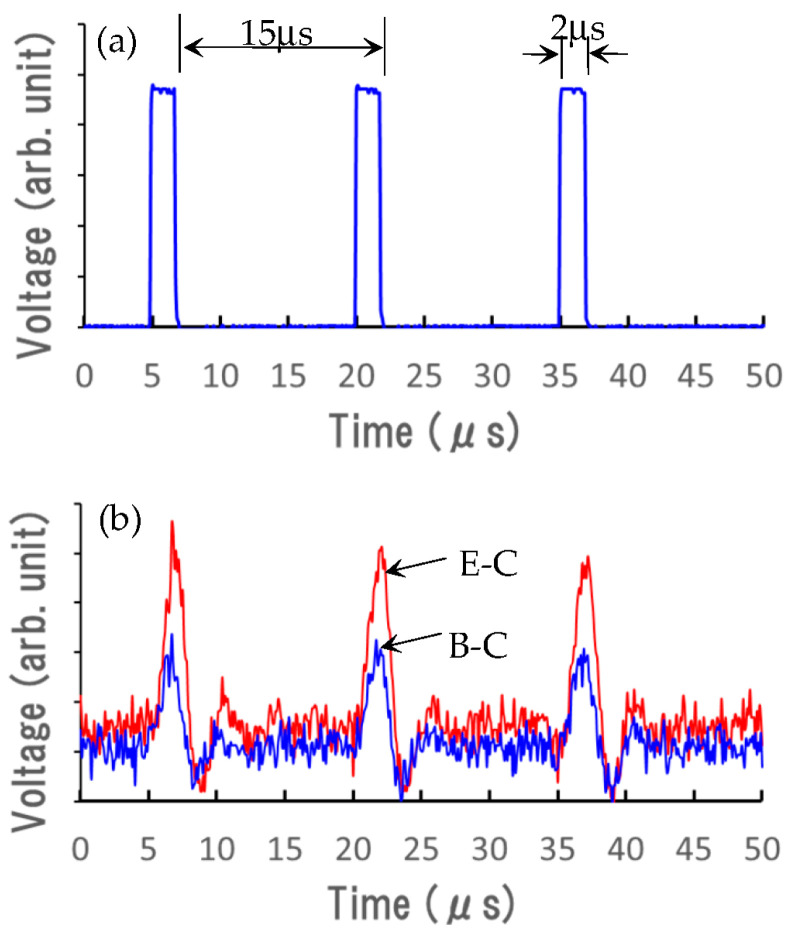
(**a**) Waveform of the photoemission from the UV illumination. (**b**) Photoresponse curves of the devices based on the β-Ga_2_O_3_/NiO/β-Ga_2_O_3_ (E–C) and the NiO/β-Ga_2_O_3_ (B–C) structures.

**Figure 9 sensors-23-08332-f009:**
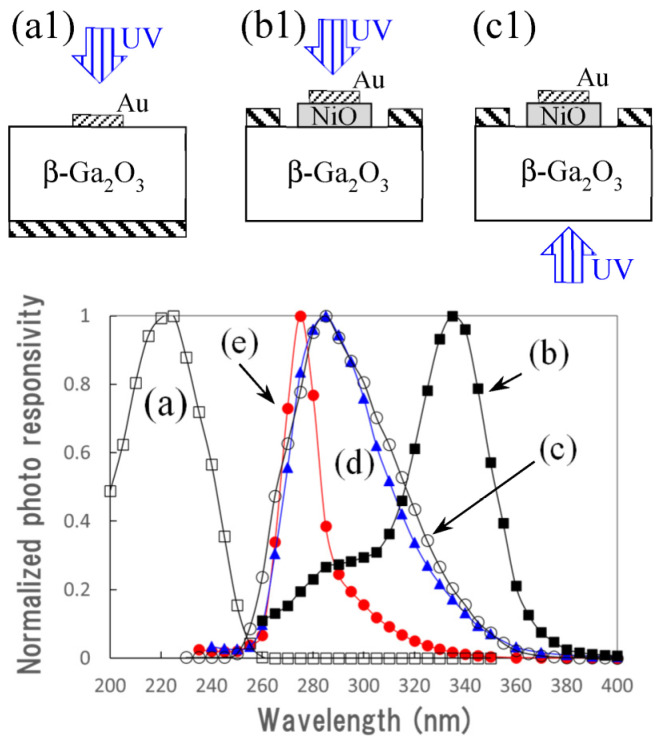
Normalized photoresponsivity of the diodes of (a) Au Schottky on β-Ga_2_O_3_, (b,c) NiO/Ga_2_O_3_ p–n diode illuminated from the surface side of NiO and the bottom side of Ga_2_O_3_ substrate, respectively. (d,e) correspond to (a,b) shown in Figure 7. (**a1**–**c1**) are diode structures corresponding to (a–c).

**Figure 10 sensors-23-08332-f010:**
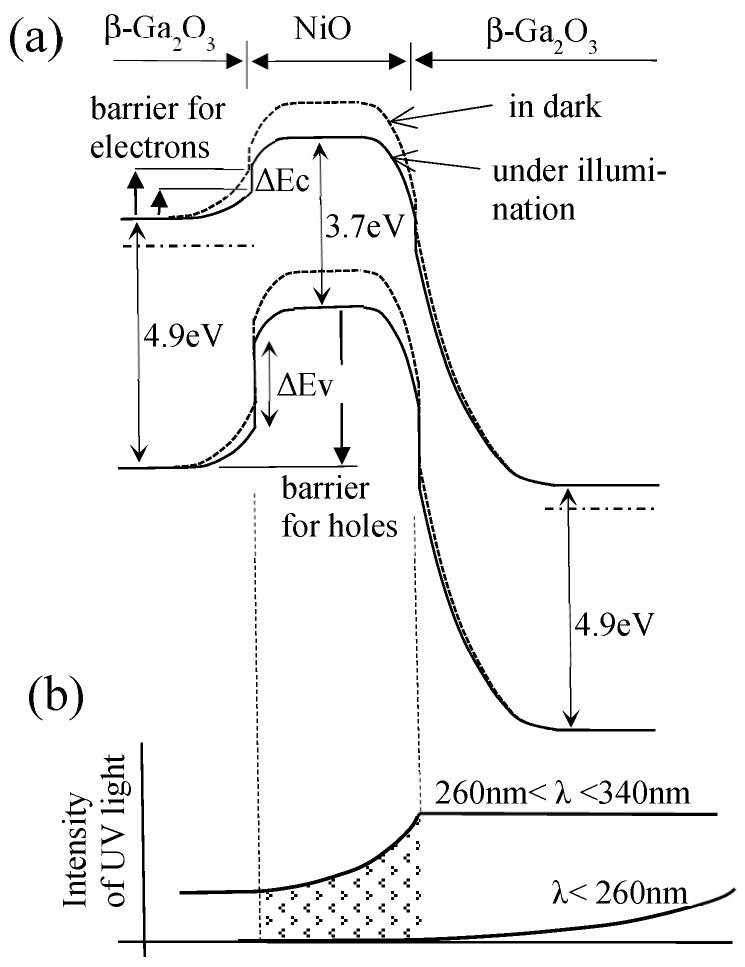
(**a**) Schematic band diagram of the device based on the β-Ga_2_O_3_/NiO/β-Ga_2_O_3_ structure. The electrode between E and C was reversely biased. (**b**) UV light absorption model.

**Table 1 sensors-23-08332-t001:** Comparison of photoresponsivity of UV photodetectors based on Ga_2_O_3_ structure.

UV Photodetectors		
Structure	Photoresponsivity	References
Transparent amorphous Ga_2_O_3_	2.66 A/W 5.78 A/W	[14] [15]
ITO/β-Ga_2_O_3_	1720.2 A/W	[16]
ITO/NiO/β-Ga_2_O_3_	27.43 A/W	[10]
β-Ga_2_O_3_/NiO	415 mA/W	[11]
β-Ga_2_O_3_/NiO with Pt nanoparticles	4.27 mA/W	[12]
β-Ga_2_O_3_/NiO	57 μA/W	[13]
β-Ga_2_O_3_NW/CH_3_NH_3_PbI_3_/NiO NW	254 mA/W	[17]
β-Ga_2_O_3_/NiO/β-Ga_2_O_3_	10 mA/W	This work

## Data Availability

The data that support this study’s results are available from the corresponding author upon request.

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
