# Peer review of "Ultraviolet Photodetector Based on a Beta-Gallium Oxide/Nickel Oxide/Beta-Gallium Oxide Heterojunction Structure"

_sensors, 2023, doi:10.3390/s23198332_

Round 1
Reviewer 1 Report
The authors synthesized an npn structure based on a β-Ga2O3-NiO/β-Ga2O3. The PD displays rectification and greater responsivity and also amplifies the photocurrent. This work provides a comprehensive understanding of UV photodetectors. I would like to recommend its publication after the authors successfully address the following questions.
1. Overall, it is a research of device performance, and there is no new understanding of intrinsic physics related to the unique properties of this device. From a view of device performance, the advantage of this detector structure is not clear.
2. The spectral performance of a detector is highly related to the absorption of the active material and the absorption of this material should be given.
3. How about the response speed of the photodetector? Response speed is an important characteristic to determine their possible applications.
4. There are recent advances on solar blind photodetector of gallium oxide, which can be added in the Introduction. (such as:J. Phys. Chem. Lett. 2023, 14, 6444; Adv. Electron. Mater., 2023, 9, 2201216; Mater. Today Phys., 2022, 28: 100883.)
5. How do the authors confirm the band alignment? X-ray photoelectron spectroscopy studies, for example, is missing.
6. To extract an exact value of dark current noise, it can be measured by Lock-in amplifier. Otherwise, the responsivity can be over-estimated. Please check it and explain it.
No
Author Response
I appreciate the reviewers for their insightful comments.
I have carefully read the comments and questions before drafting the revised manuscript and responses. Please find below my point-by-point responses to reviewers’ comments.
The additions and revisions are highlighted in yellow in the revised manuscript. Several references were also added. The revised manuscript was corrected in English by MDPI author service before submission.

Reviewer 2 Report
Recommendation: Major revisions needed as noted.
In this manuscript, the authors report on the fabrication of β-Ga2O3/NiO/β-Ga2O3 heterojunction structure and its utilization for the high-performance UV photodetection (310 nm). The device exhibited satisfactory responsivity and also, an acceptable mechanism was also proposed to explain the high performance of the UV photodetection based on heterojunction. The concept is quite good and the manuscript is also well written which will likely to attract attention in this field. However, there some severe issues that needs to be clarified and should be well addressed before this work can be finally accepted for publication in Sensors journal.
1) Please elaborate the introduction What are the benefits of p-type conduction in β-Ga2O3, over the n-type conduction in β-Ga2O3. What is the novelty of your device. I think this statement is adopted much because the theme of the work is based on heterojunction (n-p-n type). So please be assure of this. In my opinion, it should be better if the author can describe the difficulties of the β-Ga2O3, and the possible actions to overcome their limitations to active applicable for the high-performance devices. Please make the introduction much clearer.
2) Importance of UV photodetectors (PDs) should further be emphasize in great manner and compere your device structure with the other published UV PDs
3) There are many p-type semiconductors rather than NiO, like MgO, CuGaO2, NiCo2O4, Cu2O etc., Please explain in the introduction.
4) Please mention the specific UV wavelength region like UV-A, B, C like this based on the usage of the UV (310 nm) range. Authors should mention the specifications of filter. Also, please mention the UV light power density in the revised manuscript.
5) Please mention the source to the substrate distance and the area of illumination.
6) Please provide the conductivity of each sample.
7) Please provide the UV power on-off ratio and the rectification behavior under dark and UV light in the revised manuscript for better understanding of the device characteristics.
8) What is the bandgap values of the sample, since it can play a major role in UV absorption? Please provide or supplement the UV-Vis spectra.
9) In the Fig.4, 5, 6 caption the description of (b) was missed. Please check it.
10) In the photodetection properties oxygen vacancies (Ov) plays a key role in charge transport? How about Ov of β-Ga2O3/NiO? Authors needs to provide the XPS spectroscopical evidence for the samples in the revised manuscript and explained.
11) Have the authors studied the charge transfer characteristics of their heterostructure using band analysis based on XPS/UPS experiments. Because it is one of the finest ways to understand the charge generation, separation and transfer the e-h pairs. Or lese please provide the PL analysis in order to better understand the charge carriers section efficiency and carrier flow impedance.
12) How about the long-term stability?
13) Please add some more recent literatures to support your study results in Table 1.
14) However, the references cited by the author were not up to date. These latest references are closely related to this work and thus suggested to be included while discussing about the responsivity:
10.1021/acsaelm.0c00301, 10.1109/LPT.2022.3214655, 10.1002/pssr.202000518.
Author Response

(The authors gave the same response as above.)

Round 2
Reviewer 1 Report
The revised manuscript can be accepted.
English is ok.
Reviewer 2 Report
Authors have answered all the queries.
Hence this manuscript can be accepted in its current form.